# Lupus Nephritis and Dysbiosis

**DOI:** 10.3390/biomedicines11041165

**Published:** 2023-04-13

**Authors:** Marta Monticolo, Krzysztof Mucha, Bartosz Foroncewicz

**Affiliations:** 1Department of Immunology, Transplantology and Internal Diseases, Medical University of Warsaw, 02-006 Warsaw, Poland; marta.monticolo@gmail.com (M.M.); krzysztof.mucha@wum.edu.pl (K.M.); 2Institute of Biochemistry and Biophysics, Polish Academy of Sciences, 02-106 Warsaw, Poland

**Keywords:** lupus nephritis, systemic lupus erythematosus, dysbiosis, microbiome

## Abstract

Lupus nephritis (LN) is one of the most common and serious complications of systemic lupus erythematosus (SLE). The risk factors for developing LN by SLE patients are not fully understood. They are considered to be a mix of genetic and environmental variables, one of them being dysbiosis, proposed recently to interfere with autoimmunity. As of yet, the relations between the human microbiome, its genetic determinants, individual variability and clinical consequences remain to be established. One of the major obstacles in studying them is the magnitude of confounders, such as diet, drugs, infections or antibiotics use. They also make comparison between the studies extremely complicated. We reviewed the available evidence for the interplay between microbiome, dysbiosis and mechanisms triggering the autoimmune responses and potentially contributing to LN development. One such mechanism is the stimulation of autoimmune responses by bacterial metabolites that can mimic autoantigens and cause antibody production. These mimicking microbial antigens seem to be a promising target for future interventions.

## 1. Introduction

Systemic lupus erythematosus (SLE) is an inflammatory autoimmune disease with a diverse spectrum of clinical manifestations. It is characterized by a variable clinical course, presenting with systemic or organ-specific symptoms. Renal involvement, called lupus nephritis (LN), is diagnosed in more than half of SLE patients, and is associated with increased mortality [1]. The etiology of SLE and LN is very complex and may be a result of the interplay between genetic susceptibility and hormonal and environmental factors, including infections and dysbiosis [2]. Many of the known LN susceptibility genes are responsible for mediating inflammation via cytokine/chemokine production and the activation of myeloid and B cells [3]. Some of the genes are related to bacterial responses, such as the mannose-binding lectin 2 (MBL2) gene. MBL recognizes carbohydrate patterns found on the surface of numerous pathogenic microorganisms, including bacteria. Binding MBL to a bacterial pattern results in the activation of the lectin pathway of the complement system. Genetically determined MBL2 deficiency was associated with development of LN in SLE patients [4]. This association provides a rationale to study interactions between LN susceptibility genes and dysbiosis.

The human microbiome remains in a steady state thanks to the multiple controlling mechanisms, such as antibody-producing plasma cells, correct T regulatory (Treg)/helper 17 (Th17) lymphocyte ratio [5] and the anti-inflammatory cytokines and defensines secreted by jejunal epithelial cells [6]. Dysbiosis may result from a dysfunction of these controlling mechanisms, immunodeficiency associated with SLE and cellular defects. Dysbiosis may be influenced by many environmental factors, including increased consumption of antibiotics [7], non-steroidal anti-inflammatory drugs [8] and proton pump inhibitors [9]. The analysis of their use in the years 2000–2015 revealed stable increases. Interestingly, at the same time, the incidence of SLE showed a similar trend [10] (Figure 1). Immunosuppressive treatment of LN may additionally contribute to dysbiosis. The role of genetic predisposition in determining the composition of gut microbiome was also reported [11]. Dysbiosis offers potential targets for development of new therapeutic approaches. So far, probiotic supplementation and the reduction of the aforementioned environmental factors may be recommended. However, such an approach remains highly unspecific. Recognizing the role of dysbiosis in LN pathogenesis might enable more personalized interventions (Figure 2).

## 2. Microbiota

The microbiome is defined as the total number of genes of the microbiota. Available data on the microbiome and autoimmunity in humans are increasing, but still very limited. It comes mostly from the studies on the intestinal flora [12,13]. However, in recent years, it has also been observed that other bacterial reservoirs, such as the oral cavity, may be a potential source of immune response triggers that may initiate or exacerbate disease [14].

Human research methodology is different from animal research and often involves comparative studies in different diseases that may have similar pathogenesis; for example, rheumatoid arthritis or Sjogren’s syndrome [15,16]. This fact precludes us from drawing definitive conclusions. It has been shown that dysbiosis can be linked with more than 100 diseases [17], including autoimmune diseases, such as type 1 diabetes [18], rheumatoid arthritis [19] and SLE [6]. The Human Microbiome Project was established to accurately assess and catalog the human microbiota. It enabled creating a reference database and demonstrating correlations between the microbiome changes and the development of many diseases, including autoimmune diseases [20]; whereas this project involved a relatively small number of individuals, subsequent microbiome genome-wide association studies have increased sampling to overcome the small effect sizes of single variants on the composition of the gut microbiome [21]. Using 16S rRNA sequencing to characterize the gut microbiomes of healthy individuals, Wang et al. identified 42 host genetic loci that affected variability in gut microbiome composition. Interestingly, each individual locus explained only a small proportion of the total variability between different hosts, but together the loci explained 10.4% of the inter-individual gut microbiome variability [22]. Furthermore, the influence of diet on the microbiome and on the genetic susceptibility to lupus was studied in 1154 mice fed with three different diets [23]. It was observed that diet substantially contributed to the variability of complex traits and unmasked additional genetic susceptibility quantitative trait loci. Thanks to whole-genome sequencing, the candidate genes were proposed. This important study suggests that diet modifies genetic susceptibility to lupus and shifts intestinal bacterial and fungal composition, preceding clinical SLE symptoms. This study also underlines the importance of including environmental factors in genetic association studies. As an example, a study assessing host genetics in the context of gut microbiome in East Asian populations revealed that certain species, such as *Saccharibacteria* and *Klebsiella*, may be influenced by host genetics. This study also indicated that different diseases have a potential impact on the gut microbiome quality. For example, a higher risk of atrial fibrillation was associated with a lower abundance of *Lachnobacterium*, *Bacteroides Coprophilus*, *Barnesiallaceae* and a higher abundance of *Burkholderiales* and *Alcaligenceae*. The risk of prostatic cancer may be higher due to increased abundance of *Anaerostipes* and may be lowered by *Prevotella* [24]. Such associations of bacteria with organ-specific changes suggest their prognostic potential.

## 3. Microbiota, Dysbiosis and LN

In the past few years, microbiome and dysbiosis have been in the spotlight. Evidence is mounting on how the microbiota is involved in the pathogenesis of auto-immune diseases [25,26,27,28]. An association between the translocation of gut bacteria and the development of SLE was demonstrated [29,30]. Moreover, some research teams linked LN to microbiota. First, Valiente et al. demonstrated that NZM2410 mice colonized with segmented filamentous bacteria presented more severe histological lesions of LN compared to mice without dysbiosis [31]. The serum level of interleukin (IL)–17 was increased in colonized mice; however, there was no change in the amount of Th17 cells infiltrating the kidney. These last findings contrast with those observed by Mu et al., who found that the treatment of MLR/lpr lupus mice with five strains of *Lactobacillus* modified the Treg–Th17 balance in the kidney toward a Treg phenotype with an increase of IL-10 secretion [32]. In the latter report, researchers also found that lupus-prone mice supplemented with these strains showed improved renal function and survival. Finally, Azzouz et al. have shown that SLE patients, in contrast to healthy controls, presented with dysbiosis manifested as an over-abundance of anaerobic *Ruminococcus gnavus* (RG) [33]. They demonstrated that anti-RG antibodies cross-react with anti-double-stranded DNA (dsDNA) IgG and patients with active classes III and IV LN have a higher level of anti-RG antibodies in their serum. Taken together, these data highlight the association of microbiota, dysbiosis and LN.

Many studies aimed to assess the diversity of the gut microbiota and determine its “core” composition. Hevia et al. observed a reduced *Firmicutes/Bacteroidetes* ratio, two species predominantly found in the gut microbiome, and a reduced abundance of *Firmicutes phylum* bacteria in patients diagnosed with SLE [12]. In contrast, Luo et al. did not confirm these results [34]. However, their study was conducted in both men and women, while the other studies enrolled only women.

The most frequently used animal model of lupus is MRL/lpr mice with a genetic susceptibility to SLE caused by a mutation in the Fas protein coding gene. They spontaneously produce auto-antibodies and develop lupus-like symptoms [35]. However, the development of mouse lupus models depends not only on the genetic variants, but also on variable host responses to bacterial flora. It is also important to note that in different species, different forms of interferon (IFN) induce inflammation (IFNα in MRL/lpr versus IFN gamma in Toll-like receptor in BXSB) [36]. The type of activated or inhibited immune cells may also depend on the metabolites produced by the gut microbiota. Because of the magnitude of substances that influence these responses, researchers have just begun to identify these metabolites and assign particular functions to them [37].

### 3.1. Intestinal Permeability

Disturbance of the intestinal ecosystem and changes in the gut immunity may contribute to the LN development. One of its potential mechanisms may be dysbiosis-induced permeability of the intestinal barrier, leading to intestinal translocation (‘leaky gut’) (Figure 3A). A lowered *Bacteroides*/*Firmicutes* ratio plays a major role in this process [38]. In addition, *Enterococcus gallinarum* transfers from the intestine to the liver as a result of increased intestinal epithelial permeability, and stimulates dendritic cells to synthesize type 1 INF [39]. Bacterial cell wall components, such as lipopolysaccharides, enter the systemic circulation, leading to inflammation. Additionally, bacterial antigens cause the hyperactivation of lymphocytes and Tregs differentiation in the intestinal wall. As a consequence, responses against commensal microorganisms, as well as against own cells and their components, may be elicited [14]. The production of antibodies and pro-inflammatory cytokines, including IL-6, is increased, whereas the production of IgM antibodies that have a protective function under standard conditions is reduced. The inflammatory effects of dysbiosis are presented in Figure 3B,C.

Experimental studies revealed that the *Lacidobacillus* treatment inhibits uncontrolled pro-inflammatory response of T lymphocytes and LN development. Interestingly, the *Lactobacillus* supplementation decreased proteinuria and serum autoantibodies levels in LPR mice [32]. This fact may indicate a possible preventive role of probiotics in lupus development and their nephroprotective effect (Figure 3D).

### 3.2. Molecular Mimicry

One of the mechanisms by which gut microbiota may influence the development of SLE is triggering an autoimmune responses by metabolites (including peptides) secreted by microorganisms that can mimic autoantigens and cause cross-reactivity [40]. For example, it has been shown that the YLYDGRIFI peptide of the IS66 family of *Odoribacter splanchnicus* transproteases can affect IFN-γ and interleukin 17A secretion from peripheral blood mononuclear cells in a subgroup of anti-Smith (Sm) antibody-positive SLE patients. The most probable explanation for this phenomenon is that this bacterial peptide is similar to the YLYDGRIFI autoepitope of human small nuclear ribonucleoprotein-associated proteins B and B’, an antigen presented to T cells by the Human Leukocyte Antigen (HLA)—DR isotype. Moreover, the DGQFCM peptide, which is derived from *Akkermansia miciniphila* (present in higher titers in SLE patients), mimics the extracellular DGQFCG part of the human Fas ligand and could affect the binding to IgG produced by memory B cells in a subset of SLE patients [39] (Figure 3D). Accordingly, Zhang et al. reported that purified bacterial antigens of *Burkholderia* and transcriptional regulatory peptide referred to as RAGTDEGFG are able to bind to serum dsDNA antibodies in SLE patients. These findings may suggest that the formation of anti-dsDNA antibodies may involve molecular mimicry of *Burkholderia* [41]. This is of particular importance for potential renal involvement in SLE patients. It is well known that anti-dsDNA antibodies act as a diagnostic marker for LN. Anti-dsDNA have been implicated in the pathogenesis of LN; they are present in higher concentrations in renal tissue compared to systemic circulation and their increases in serum may precede lupus flares. Moreover, pre-emptive therapy based on the rising anti-dsDNA antibody levels can prevent LN flares [42]. For these reasons, the fluctuations in anti-dsDNA titer are widely used to monitor disease activity [43]. The evidence for the link between bacterial antigens and anti-dsDNA antibodies formation was previously demonstrated by Y. Schoenfeld et al., who found that mycobacterial cell wall glycolipids were associated with anti-dsDNA autoantibodies both in SLE patients and in mouse models [44]. Of course, the interaction between tuberculosis and SLE is complex, as one seems to be a risk factor for the development of the other. However, the glycolipid molecular mimicry may partially explain it.

Taking into account that the HLA-DR region (particularly the DR3 allele) is the dominant lupus susceptibility locus and the importance of T cells in SLE development, Zhao et al. showed multiple intramolecular DR3 restricted T cell epitopes in the Sm D protein, from which they generated a non-homologous, bacterial epitope mimicry library. From this library they identified ABC247-261 as one new DR3 restricted bacterial T cell epitope that mimics the ABC transporter ATP-binding protein in *Clostridium tetani*. It activated and induced autoreactive SmD66-80-specific T cells and induced synthesis of autoantibodies to lupus-related autoantigens in vivo. Thus, their group provided grounds for further research on the mimicry of bacterial epitopes, which could lead to autoimmunity in susceptible DR3 individuals [45].

The Ro/La system is considered a heterogeneous antigenic complex, consisting of different proteins, including Ro60 (60 kDa). Anti-Ro60 antibodies are part of the family of anti-Ro/SSA antibodies, historically markers of SLE and the most frequently encountered autoantibodies in patients with connective tissue diseases. The cross-reactivity between Epstein–Barr virus nuclear antigen 1 (EBNA-1) and Ro60 proteins has been suggested as a possible mechanism for the anti-Ro60 antibody response both in the rabbit model and in patients with SLE. Szymula et al. focused on the mimicry ability of the peptides from oral, skin and intestinal bacteria to activate T cell hybrids reacting with Ro60. They demonstrated that cross-reactive B cells which recognize EBNA1 peptide 58–72 and Ro60 peptide 169–180 are involved in the initiation of anti-Ro60 antibody responses in SLE patients. They also confirmed the significant role of the HLA-DR3 region in response to Ro60 [46].

Such mechanisms of molecular mimicry provide a background for therapeutic interventions. Bacteria or bacterial epitopes could be manipulated to prevent the induction of autoimmune responses. Therefore, the identification of specific bacterial species and peptides could become an attractive research target.

Another link between intestinal dysbiosis and immune responses was reported by Lopez et al., who demonstrated in vitro that fecal samples isolated from patients with active lupus promote lymphocyte activation and differentiation of naïve CD4+ to Th17 lymphocytes [47]. Changing the balance between Th17 and Tregs may lead to LN exacerbation, and can also increase antibody production and deposition in the kidneys, whereas enrichment of SLE stool samples with Treg-inducing bacteria, such as a mixture of two *Clostridia* strains, showed that they significantly reduced the Th17/Th1 balance. Interestingly, the *Bifidobacterium bifidum* supplementation was able to prevent CD4(+) lymphocyte over-activation. These findings may support the potential therapeutic benefit of probiotics containing Treg-inducing strains in order to restore the Treg/Th17/Th1 imbalance present in SLE patients [47].

It is very important to remember that dysbiosis in LN patients may result both from the disease and its therapy. For example, steroid therapy leads to a significant change in the intestinal microflora in comparison to healthy individuals. The group of patients taking steroids increased the number of *Firmicutes* versus *Bacteroides* strains and also had a significantly higher Faith’s phylogenetic diversity compared to healthy controls and patients without steroid therapy [47]. Although the molecular patterns remain unknown, twelve cytokines showed higher expression levels. Interestingly, the expression of IFN-γ, IL-2, IL-10, IL-35 and tumor necrosis factor was significantly higher in patients treated with steroids. The immunosuppression (IS), regardless of the type, reduces both the abundance and diversity of bacteria in SLE patients. Moreover, IS increases the risk of bacterial infections, whereas antibiotics used to control them have additional impact on dysbiosis [47]. Furthermore, long-term IS or steroids use may increase the frequency of gastrointestinal complications and need for PPI use. These drugs also have the potential to change the microbiome [48]. These multiple confounders make studying dysbiosis in patients with SLE very difficult and comparison between the studies extremely complicated [47].

## 4. Oral Microbiota

Until now, intestinal microflora has been the preferred flora to analyze, probably because of the abundance of species, as well as the relatively easy method of obtaining material for analysis. However, the oral microbiota may also cause or exacerbate the disease, and some oral bacteria metabolites can interfere with disease-related genes. Recent studies showed the potential association of oral microbial DNA with nucleic acid sensing, suggesting a link between oral and systemic diseases, including SLE [49].

The oral cavity is an important reservoir of bacteria. However, its composition is highly variable, mainly due to exposure to numerous external and internal factors, such as smoking, food, poor hygiene, periodontitis, or salivary disorders. In a healthy individual, the oral microflora remains in symbiosis with its host. The dysbiosis caused by microflora imbalance may lead to periodontal tissue damage and increased systemic inflammation. In the long term, it can increase the risk of other diseases, including SLE. Importantly, one manifestation of lupus is oral mucosal lesions, occurring in 5–40% of patients. This high prevalence of mucosal lesions in SLE may indicate that the oral microflora may play a role in the development of oral symptoms. It may also contribute to systemic exacerbations through the production of autoantibodies against products of the oral microflora [50].

Correa et al. showed that the oral flora in SLE patients is significantly different from flora in healthy individuals [51]. Even if they have not developed an inflammatory process, the flora are more abundant, including species such as *Fretibacterium*, *Prevotella* and *Selenomonas*. Periodontitis is also much more common in patients with SLE, begins at a younger age and is exacerbated in patients with other co-existing infections [52].

These findings were supported by Pessoa et al., who showed a significant increase in cytokines (especially IL-1 β and IL-6) and development of inflammation of oral mucosa in patients with autoimmune diseases. The authors suggested that subgingival bacterial species associated with SLE have a significant impact on the host systemic cytokine system, influencing general condition [53]. Importantly, Bei-di Chen et al. proved that many species present in the gut (strains of *Shuttleworthia satelles*, *Actinomyces massiliensis*, *Clostridium Species ATCC BAA-442*, *Bacteroides fragilis* or *Clostridium leptum*) originate partially from the oral microbiota [54].

## 5. Genital Microbiota

Variability in the microflora of other environments, such as the upper respiratory tract or genital tract, may correlate and influence autoimmune responses and the development of SLE. However, additional studies are necessary to establish the role of these “niche” dysbiosis in SLE development. Although lupus affects women significantly more often than men, there are not many studies linking its development to uro-genital dysbiosis or infections. It is known that the microflora present in the genital tract changes with time. These differences may be caused by natural hormonal changes related to puberty and the beginning of menstruation. The changes may be also caused by the initiation of sexual activity, and are constantly modified throughout a woman’s life until menopause. During this period, SLE development is most frequent.

The vaginal dysbiosis in SLE patients may be influenced both by the disease and by its treatment. Mendez-Martinez et al. conducted a survey for the most common pathogenic species in the genital tract in Mexican women, which is the *Mycoplasma* species. They found that *Ureaplasma urealyticum* was the only pathogen found in SLE patients. However, it might have been related to taking steroids (as many as 81.5% of subjects were treated with prednisone) [55]. Ekiel et al., in their preliminary study of the Polish population, showed that in women with active SLE there are no significant differences in the occurrence of mycoplasmas compared to healthy patients, but statistically the most frequent species was *U. parvum* [56].

It is known that the risk of urinary tract infections is significantly higher in women. In addition, in female patients with SLE, this complication is a crucial determinant of the course of the underlying disease and its exacerbations. Rotjanapan et al., in a pilot study of 52 patients with SLE, showed that the microflora of the urinary tract of patients with active infection is significantly different compared to that of healthy patients. They also documented that the use of prophylactic antibiotic therapy with trimethoprim sulfamethoxazole had an impact on the vaginal microflora. A relationship like this can favor the development of pathogenic bacteria, resulting in an infection and exacerbation of SLE [57].

Unfortunately, the data on genital dysbiosis and lupus nephritis are lacking.

## 6. Skin Microbiota

The skin is one of the most common body barriers. It determines our contact with the environment and is an important protection against external pathogens. Up to 80% of patients with SLE present symptoms of the disease expressed as skin lesions, and in 25% of them skin lesions are the first manifestation of the disease [58]. Zhou et al. revealed that the skin microflora of SLE patients presents significant differences when compared to healthy controls. Importantly, there are apparent species changes on skin with rash compared to skin without lesions—*Curvibacter* were decreased in rash region compared to non-rash region and the genera *Pelagibacterium* and *Novosphingobium* were increased [59]. Interestingly, a comparison between remission and active SLE groups revealed that the family *Caulobacteraceae* was positively correlated with SLE disease activity index (SLEDAI) and negatively with complement C3. Additionally, *Aerococcaceae* was negatively correlated with SLEDAI and immunoglobulin G [58]. These observations provided some suggestive evidence for further exploration of skin microbiota in SLE patients.

## 7. Conclusions

Genetic predisposition is one of the underlying causes of dysbiosis. However, dysbiosis may change over time depending on multiple endocrine and environmental factors. Current approaches based on the modifications of nutrition and regulation of certain medication intake may be important in reducing the risk for LN development and exacerbations, but are very untailored. It seems that the presence of dysbiosis can cause the development of the disease, and its variability at different levels modifies cellular responses. Combined with genetic and environmental predispositions, it may have a significant impact on disease development. The bacterial peptides identified so far that mimic autoantibody production should be a starting point for further identification of disease triggers. They could become personalized targets for future interventions.

Effective and complete identification of the basic, “healthy” microbiome and the captured moments that lead directly to its changes remain the most challenging problem. The methodology for studying the precise course of its modification also remains undefined and problematic.

Common methods of bacterial identification, such as Next-Generation Sequencing, bioinformatics genes mapping and the advancements of these methods can be of great help in the diagnosis of causes due to their diversity, multipotentiality and huge variability; however, the characterization of target points is very difficult and requires a lot of resources and trials.

The approach of the future remains direct personalized therapeutic management, in addition to the possibility of creating antibacterial vaccines that would protect at-risk patients from adverse changes in the microbiome, resulting in destructive changes and leading to SLE and LN development.

## Figures and Tables

**Figure 1 biomedicines-11-01165-f001:**
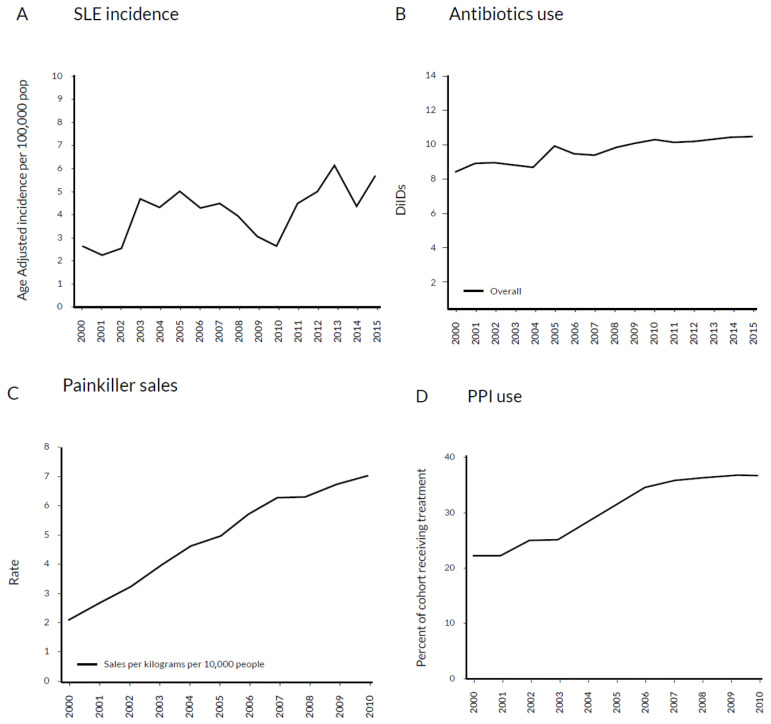
Similarities in trends of increased SLE incidence and consumption of antibiotics, nonsteroidal anti-inflammatory drugs and proton pump inhibitors in the years 2000–2015. (**A**)Age-adjusted systemic lupus erythematosus incidence per 100,000 population [10]. (**B**) Antibiotic consumption rate in daily doses [7]. (**C**) Nonsteroidal anti-inflammatory drug sales in kilograms per 10,000 people [8]. (**D**) Proton pump inhibitor consumption [9]. DDs—daily doses; DDD—defined daily dose; PPI—proton pump inhibitor.

**Figure 2 biomedicines-11-01165-f002:**
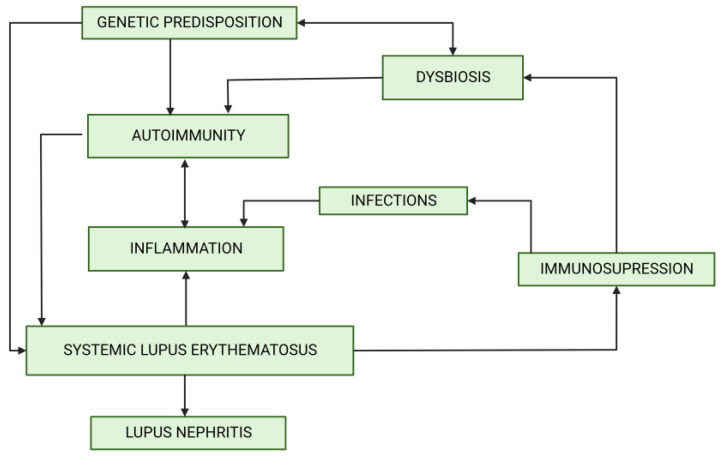
The associations between dysbiosis and factors involved in lupus nephritis development. Created by BioRender.com.

**Figure 3 biomedicines-11-01165-f003:**
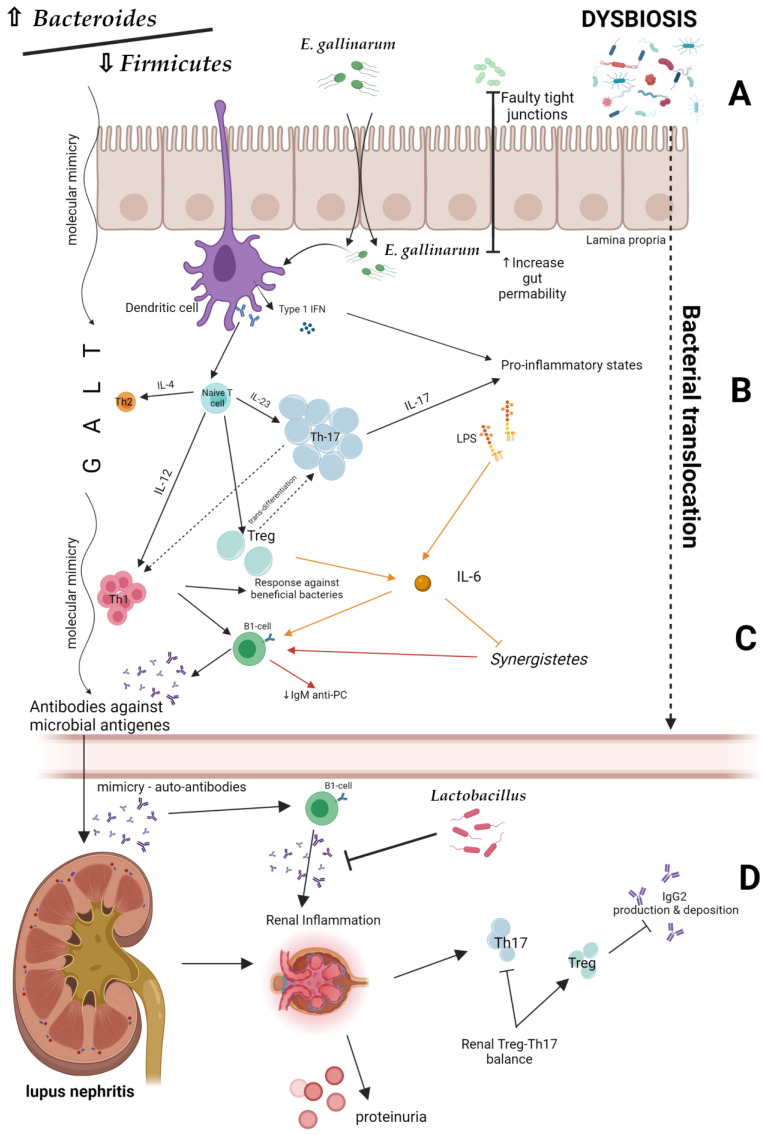
Potential mechanisms of lupus nephritis development [22,34,38,39]. Created by BioRender.com. (**A**) Intestinal barrier and bacterial translocation. (**B**,**C**) The inflammatory effects of dysbiosis. (**D**) Possible preventive role of probiotics in lupus nephritis development. IL—interleukin; IgG2—Immunoglobulin G2; Th—T helper; IgM anti-PC—Immunoglobulin M anti-phosphorylcholine; GALT—gut-associated lymphoid tissue.

## Data Availability

Not applicable.

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
