# Peer review of "Lupus Nephritis and Dysbiosis"

_biomedicines, 2023, doi:10.3390/biomedicines11041165_

Round 1

Reviewer 1 Report

This topic has been well reviewed recently which can be searched with Pubmed, and very few orinial studies were reported, and no novel fings recently.

Also, as the author stated, the risk factors for developing LN by SLE patients are not fully understood, especially for the role of dysbiosis in LN. 

So, in my opinion, the part of conclusions need to be reconsidered, for examlple: "Obviously, dysbiosis itself is not responsible for disease development.", "The next step in linking dysbiosis to LN would be identification of microbial peptides that mimick antigens that initiate autoantibodies production."

Reviewer 2 Report

The article is consistent within itself. The references are relevant and recent. The cited sources are referenced correctly. Appropriate and key studies are included. The paper is comprehensive, the flow is logical and the data is presented critically.

However, there are some specific comments on weaknesses of the article and what could be improved:

Major points - none

Minor points

1. What is the search strategy for this review - which databases did you use, which keywords and combination.

2. What is still a challenge - please, add at the end.

Round 2

Reviewer 1 Report

For your response to reviewers, "In our research, we mainly used combinations of the words "lupus nephritis" and "dysbiosis"; "dysbiosis" and "genetic predisposition"; "microbiome" and "lupus nephritis."

In addition to "lupus nephritis", "Systemic Lupus Erythematosus" is also suggested to added to the key words, these two words cannot be seperately clearly. Then, you will find other close related references for this topic, e.g. Gut Microbiota Dysbiosis in Systemic Lupus Erythematosus: Novel Insights into Mechanisms and Promising Therapeutic Strategies. Front Immunol. 2021 Dec 3;12:799788.
